# Acquiring Target Stacking Skills by Goal-Parameterized Deep Reinforcement Learning

## Abstract

Understanding physical phenomena is a key component of human intelligence and enables physical interaction with previously unseen environments. In this paper, we study how an artificial agent can autonomously acquire this intuition through interaction with the environment. We created a synthetic block stacking environment with physics simulation in which the agent can learn a policy end-to-end through trial and error. Thereby, we bypass to explicitly model physical knowledge within the policy. We are specifically interested in tasks that require the agent to reach a given goal state that may be different for every new trial. To this end, we propose a deep reinforcement learning framework that learns policies for stacking tasks which are parametrized by a target structure – departing from conventional approaches based on simulation and planning. We validated the model on a toy example navigating in a grid world with different target positions and in a block stacking task with different target structures of the final tower. In contrast to prior work, our policies show better generalization across different goals.

## 1 Introduction

Understanding and predicting physical phenomena in daily life is an important component of human intelligence. This ability enables us to effortlessly manipulate objects in unseen conditions. It is an open question how this kind of knowledge can be represented and what kind of models could explain human manipulation behavior (Yildirim et al., 2017). In this paper we are concerned with the question of how an artificial agent can autonomously acquire physical interaction skills through trial and error.

Until recently, researcher have attempted to build computational models for capturing the essence of physical events via machine learning methods from sensory inputs (Mottaghi et al., 2016; Wu et al., 2015; Fragkiadaki et al., 2016; Bhattacharyya et al., 2018; Lerer et al., 2016; Li et al., 2016). Yet there is little work to investigate how the knowledge captured by such a model can be directly applied for manipulation.

In this work, we aim to learn block stacking through trial-and-error, bypassing to explicitly model the corresponding physics knowledge. For this purpose, we build a synthetic environment with physics simulation, where the agent can move and stack blocks and observe the different outcomes of its actions. We apply deep reinforcement learning to directly acquire the block stacking skill in an end-to-end fashion.

While previous work focuses on learning policies for a fixed task, we introduce goal-parameterized policies that facilitate generalization of the learned skill to different targets. We study this problem in the aforementioned block stacking task in which the agent has to reproduce a tower as shown in an image. The agent has to stack blocks into the same shape while retaining physical stability and avoiding pre-mature collisions with the existing structure.

In particular, we aim to learn a single model to guide the agent to build different shapes on request. This is generally not intended in conventional reinforcement learning formulations where the policy is typically optimized to reach a specific goal. In our learning framework, the varying goals are given to the agent as input. We first validated this extended model on a toy example where the agent has to navigate in a gridworld. Both, the location of the start and end point are randomized for each episode. We observed good generalization performance.

Then we apply the framework to the block stacking task. We show that execution depends on the desired target structure and observe promising results for generalization across different goals.

## 2    RELATED WORK

Humans possess the amazing ability to perceive and understand ubiquitous physical phenomena occurring in their daily life. There is research in psychology that seeks to understand how this ability develops. Baillargeon (2002) suggest that infants acquire the knowledge of physical events at a very young age by observing those events, including support events and others. Interestingly, in a recent work Denil et al. (2017), the authors introduce a basic set of tasks that require the learning agent to estimate physical properties (mass and cohesion combinations) of objects in an interactive simulated environment and find that it can learn to perform the experiments strategically to discover such hidden properties in analogy to human's development of physics knowledge.

Battaglia et al. (2013) proposes an intuitive physics simulation engine as an internal mechanism for such type of ability and found close correlation between its behavior patterns and human subjects' on several psychological tasks.

More recently, there is an increasing interest in equipping artificial agents with such an ability by letting them learn physical concepts from visual data. Mottaghi et al. (2016) aim at understanding dynamic events governed by laws of Newtonian physics and use proto-typical motion scenarios as exemplars. Fragkiadaki et al. (2016) analyze billiard table scenarios and learn dynamics from observation with explicit object notion. An alternative approach based on boundary extrapolation Bhattacharyya et al. (2018) addresses similar settings without imposing any object notion. Wu et al. (2015) aims to understand physical properties of objects based on explicit physical simulation. Mottaghi et al. (2017) proposes to reason about containers and the behavior of the liquids inside them from a single RGB image.

Moreover, Lerer et al. (2016) propose using a visual model to predict stability and falling trajectories for simple 4 block scenes. Li et al. (2016) investigate if and how the prediction performance of such image-based models changes when trained on block stacking scenes with larger variety. They further examine how the human's prediction adapts to the variation in the generated scenes and compare to the learned visual model. Each work requires significant amounts of simulated, physically-realistic data to train the large-capacity, deep models.

Another interesting question that has been explored in psychology is how knowledge about physical events affects and guides human's actual interaction with objects Yildirim et al. (2017). Yet it is not clear how machine model trained for physics understanding can be directly applied into real-world interactions with object and accomplish manipulation tasks. Li et al. (2017) makes a first attempt along this direction by extending their previous work (Li et al., 2016) on stability classification. They task a robot to place a wooden block on an existing structure while maintaining stability. Placement candidates are first generated and then evaluated through the visual stability classifier, so that only predicted stable placements are executed on the robot.

In this paper, reinforcement learning is used to learn an end-to-end model directly from the experience collected during interaction with a physically-realistic environment. The majority of work in reinforcement learning focuses on solving task with a single goal. However, there are also tasks where the goal may change for every trial. It is not obvious how to directly apply the model learned towards a specific goal to a different one. An early idea has been proposed by Kaelbling (1993) for a maze navigation problem in which the goal changes. The author introduces an analogous formulation to the Q-learning by using shortest path in replacement of the value functions. Yet there are two major limitations for the framework: 1) it is only formulated in tabular form which is not practical for application with complex states 2) the introduced shortest path is very specific to the maze navigation setting and hence cannot be easily adapt to handle task like target stacking. In contrast, we propose a goal-parameterized model to integrate goal information into a general learning-based framework that facilitates generalization across different goals. The model has been shown to work on both a navigation task and target stacking.

Notably, Schaul et al. (2015) also propose to integrate goal information into learning. However, they learn an embedding for state and goal to allow generalization in a reinforcement learning setup. The process is then incorporated into the Horde framework Sutton et al. (2011), where each agent

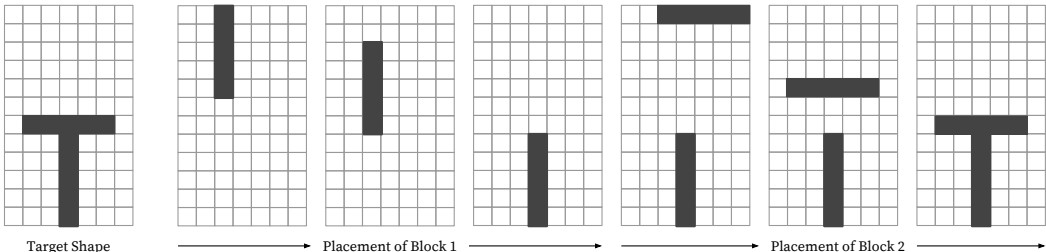

Figure 1: Target stacking: Given a target shape image, the agent is required to move and stack blocks to reproduce it.

learns towards different goals. In our work, we do not introduce a dedicated embedding learning but instead resort to an end-to-end approach where the function approximator will learn a direct mapping from sensory observations to actions that allows generalization across different goals. In addition, our work is the first to bring this concept to bear towards manipulation and planing under physical constraints – breaking with more conventional simulation and planning approaches (e.g. Yildirim et al. (2017)).

## 3 TARGET STACKING TASK

We introduce a new manipulation task: *target stacking*. In this task, an image of a target structure made of stacked blocks is provided. Given the same number of blocks as in the target structure, the goal is to reproduce the structure shown in the image. The manipulation primitives in this task include moving and placing blocks. This is inspired by the scenario where young children learn to stack blocks to different shapes given an example structure. We want to explore how an artificial agent can acquire such a skill through trial and error.

### 3.1 TASK DESCRIPTION

For each task instance, a target structure is generated and its image is provided to the agent along with the number of blocks. Each of these blocks has a fixed orientation. The sequence of block orientations is such that reproducing the target is feasible. The agent attempts to construct the target structure by placing the blocks in the given sequence. The spawning location for each block is randomized along the top boundary of the environment. A sample task instance is shown in Figure 1.

### 3.2 TASK DISTINCTION

The following characteristics distinguish this task from other tasks commonly used in the literature.

**Goal-Specific** A widely-used benchmark for deep reinforcement learning algorithm are the Atari games (Bellemare et al., 2013) that were made popular by Mnih et al. (2013). While this game collection has a large variety, the games are defined by a single goal or no specific goal is enforced at a particular point in time. For example in Breakout, the player tries to bounce off as many bricks as possible. In Enduro, the player tries to pass as many cars as possible while simultaneously avoiding cars.

In the target stacking task, each task instance differs in the specific goal (the target structure), and all the moves are planned towards this goal. Given the same state, moves that were optimal in one task instance are unlikely to be optimal in another task instance with a different target structure. This is in contrast to games where one type of move will most likely work in similar scenes. This argument also applies to AI research platforms with richer visuals like VizDoom (Kempka et al., 2016).

**Longer sequences** Target stacking requires looking ahead over a longer time horizon to simultaneously ensure stability and similarity to the target structure. This is different from learning to poke (Agrawal et al., 2016) where the objective is to select a motion primitive that is the optimal next



Figure 2: Example scenes constructed by the learned agent.

action. It is also different from the work by Li et al. (2017) that reasons about the placement of one block.

**Rich Physics Bounded**   Besides stacking to the assigned target shape the agent needs to learn to move the block without colliding with the environment and existing structure and to choose the block's placement wisely not to collapse the current structure. The agent has no prior knowledge of this. It needs to learn everything from scratch by observing the consequence (collision, collapse) of its actions.

### 3.3    ENVIRONMENT IMPLEMENTATION

A deep reinforcement learning agent requires to learn from a larger number of samples. To enable this, we build a simulated environment for the agent to interact with physical-realistic task instances. While we keep the essential parts of the task, at its current stage the simulated environment remains an abstraction of a real-world robotics scenario. This generally requires an integration of multiple modules for a full-fledged working system, such as Toussaint et al. (2010), which is out of scope of this paper.

In detail, the simulated stacking environment is implemented in Panda3D (Goslin & Mine, 2004) with bullet (Coumans, 2010) as physics engine. The block size follows a ratio of $l : w : h = 5 : 2 : 1$, where $l,w,h$ denote length, width and height respectively. We ignore the impact during block placement and focus on the resulting stability of the entire structure. Once the block makes contact with the existing structure, it is treated as releasing the block for a placement. In each episode, if the moving block collides with the environment boundary or existing structure, it will terminate the current episode. Further, if the block placement causes the resulting new structure to collapse, it will also end the episode. Stability is simulated similar to Li et al. (2017) by comparing the change of displacement across all the blocks to a pre-selected small threshold within a fraction of time. If all of the blocks' displacements are below this threshold, the structure is deemed stable, otherwise unstable. To simplify the setting, we further constrain the action to be {left, right, down}.

The physics simulation runs at $60Hz$. However considering the cost of simulation we only use it when there is contact between the moving block and the boundary or the existing structure. Otherwise, the current block is moving without actual physics simulation. To further reduce the appearance difference caused by varying perspective, the environment is rendered using orthographic projection. Figure 2 shows example images. The environment provides a user-friendly Python interface (similar to Gym(Brockman et al., 2016)) so that it can be used to test different reinforcement learning agents. At time of publication we will release our implementation of the environment.

## 4    GOAL-PARAMETERIZED DEEP Q NETWORKS (GDQN)

As one major characteristic of this task is that it requires goal-specific planning: given the same or similar states under different objectives, the optimal move can be different. To this end, we extend the typical reinforcement learning formulation to incorporate additional goal information.

## 4.1 LEARNING FRAMEWORK

In a typical reinforcement learning setting, the agent interacts with the environment at time $t$, observes the state $s_t$, takes action $a_t$, receives reward $r_t$ and transits to a new state $s_{t+1}$. A common goal for a reinforcement learning agent is to maximize the cumulative reward. This is commonly formalized in form of a value function as the expected sum of rewards from a state $s$, $\mathbf{E}[\sum_{i=0}^{T} \gamma^i r_{t+i+1}|s_t = s, \pi]$ when actions are taken with respect to a policy $\pi(a|s)$, with $0 \leq \gamma \leq 1$ being the discount factor, T for the final time step. The alternative formulation to this is the action-value function $Q^\pi(s,a) = \mathbf{E}[\sum_{i=0}^{T} \gamma^i r_{t+i+1}|s_t = s, a_t = a]$.

Value-based reinforcement learning algorithms, such as Q-learning (Watkins & Dayan, 1992) directly search for optimal Q-value function. Recently by incorporating deep neural network as a function approximator for $Q$-function, the DQN (Mnih et al., 2015) has shown impressive results across a variety of Atari games.

**DQN** For our task, we apply a *Deep Q Network* (DQN) which uses a deep neural network for approximating the action-value function $Q(s, a; \theta)$, mapping from an input state $s$ and action $a$ to Q values. In particular, two important improvements have been proposed by Mnih et al. (2015) for the learning process, including (1) experience replay, the agent stores observed transitions in a memory buffer for some time, and uniformly samples from the memory to update the network (2) the target network, agent maintains two networks for the loss function — one for the current estimator of Q function and one for the surrogate of the true Q function. For the current estimator, the parameters are constantly updated. For the surrogate, the parameters are only updated for every certain number of steps from the current estimator network otherwise kept fixed.

**Learning Goal-Parameterized Policies** To plan with respect to the specific goal, we can parametrize the agent's policy $\pi$ by the goal $g$:

$$\pi(s, g, a) \tag{1}$$

Since in this work, we applies DQN as value-based method, this corresponds to the update to original Q function with the additional goal information. The new Q-value function is hence defined as:

$$Q^\pi(s, g, a) = \mathbf{E}[\sum_{i=0}^{T} \gamma^i r_{t+i+1}|s_t = s, g, a_t = a] \tag{2}$$

As shown in Figure 3, in contrast to the original DQN model, where state and action are used to estimate Q-value, the new model further include the current goal into the network to produce the estimate. We call this model as Goal-Parametrized Q Network (GDQN).

The resulted loss function is as:

$$L_Q = \mathbf{E}[(R + \gamma \max_a' Q^\pi(s', g, a'; \theta^-) - Q(s, g, a; \theta))^2] \tag{3}$$

where $\theta^-$ are the previous parameters and the optimization is with respect to $\theta$.

## 4.2 IMPLEMENTATION DETAILS

The DQN agent is implemented in Theano and Keras to adapt to the settings in our experiment, while we use a 2 hidden layer (each with $64$ hidden units and rectified linear activation) multilayer perceptron (MLP) for most cases, we additionally swap the MLP with the CNN and follow the reported parameter settings as in the original paper (Mnih et al., 2015) to ensure our implementation can reach similar performance.

Note we don't apply the frame-skipping technique (Bellemare et al., 2012) used for Atari games (Mnih et al., 2015) allowing the agent sees and selects actions on every $k$th frame where its last

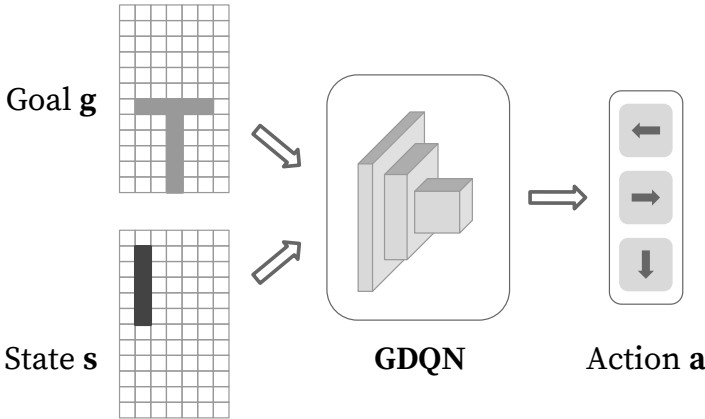

Figure 3: Our proposed model GDQN which extends the $Q$-function approximator to integrate goal information.

action is repeated on skipped frames. It does not suit our task, in particular when the moving block is getting close to the existing structure, simply repeating action decided from previous frame can cause unintended collision or collapse.

**Reward**    In the target stacking task, the agent gets reward $+1$ when the episode ends with complete reproduction of the target structure, otherwise $0$ reward.

Further, we explore reward shaping (Ng et al., 1999) in the task providing more prompt intermediate reward. Two types of reward shaping are included: overlap ratio and distance transform.

For the overlap ratio, for each state $s_t$ under the same target $g_i$, an overlap ratio is counted as the ratio between the intersected foreground region (of the current state and the target state) and the target foreground region (shown in Figure 4a):

$$o(s_t, g_i) = \frac{s_t \cap g_i}{g_i} \tag{4}$$

For each transition $(s_t, a_t, s_{t+1})$, the reward is defined by the change of overlap ratio before and after the action:

$$r_t = \begin{cases} 1, & \text{if } \Delta o_{t \to t+1} = o(s_{t+1}) - o(s_t) > 0 \\ -1, & \text{if } \Delta o_{t \to t+1} = o(s_{t+1}) - o(s_t) < 0 \\ 0, & \text{otherwise} \end{cases} \tag{5}$$

The intuition is that actions increasing the current state to become more overlapped with the target scene should be encouraged.

For the distance transform (Fabbri et al., 2008), it generates a map $D$ whose value in each pixel $p$ is the smallest distance from it to a target object $O$:

$$D(p) = \min\{\text{dist}(p, q) | q \in O\} \tag{6}$$

where dist can be any valid distance metric, like Euclidean or Manhattan distance.

For each state $s_t$ under the same target $g_i$, a distance to the goal is the sum of all the element-wise distance in $s_t$ to $g_i$ under $D_{g_i}$ (shown in Figure 4b) as:

$$d(s_t, g_i) = \sum_j D_{g_i}(s_t^j), s_t^j \in s_t \tag{7}$$

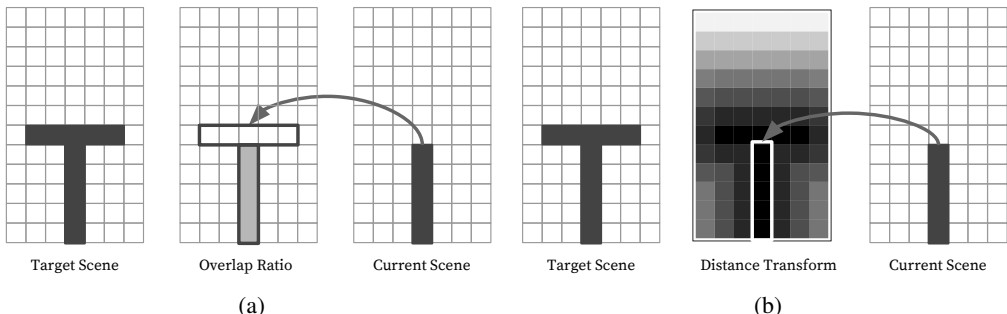

Figure 4: Reward shaping used in target stacking. (a): overlap ratio to the target. The gray area in the middle figure denotes the intersected foreground region between current and target scene, and the overlap ratio is the ratio between the areas of the two. (b): distance under the distance transform of the target. The middle figure denotes the distance transform under the target shown in the left. The distance from current scene to the target is the sum of distances masked by the current scene in the distance transform.

For each transition $(s_t, a_t, s_{t+1})$, the reward is defined as:

$$r_t = \begin{cases} 1, & \text{if } \Delta d_{t \to t+1} = d(s_{t+1}) - d(s_t) < 0 \\ -1, & \text{if } \Delta d_{t \to t+1} = d(s_{t+1}) - d(s_t) > 0 \\ 0, & \text{otherwise} \end{cases} \tag{8}$$

The intuition behind this is that action decreasing the distance between the current state and the target scene should be encouraged.

## 5 EXPERIMENTS

We evaluate the proposed GDQN model on both a navigation task and target stacking and compare it to the base DQN model which does not integrate goal information. In addition, we include the result from GDQN model with different ways of reward shaping in the target stacking task.

### 5.1 TOY EXAMPLE WITH GOAL INTEGRATION

As a toy example, we introduce a type of navigation task in the classic gridworld environment. The locations for the starting point and goal are randomized for each episode. The agent needs to reach the goal with four possible actions $\{\text{left}, \text{right}, \text{up}, \text{down}\}$ as shown in Figure 5a. Action that will make the agent go off the grid will leave it stay in the same location. The episode only terminates once the agent reaches the goal. The agent only receive reward $+1$ when reaching the current goal. Two different sizes of gridworld are tested at $5 \times 5$ and $7 \times 7$.

The training epoch size is 1000 in steps for the smaller gridworld and 3000 for the larger one, the test sizes are the same for both at 100. All the agents run for 100 epochs and the $\epsilon$ for $\epsilon$-greedy anneals linearly from 1.0 to 0.1 over the first 20 epochs, and fixed at 0.1 thereafter. The memory buffer size is set the same to the annealing length, i.e. for the smaller gridworld, the buffer size equals to the length 20 epochs in training with 20000 steps whereas for the larger one, the buffer size is 30000 steps. We measure the proportion of episodes in the test epoch that reaches the goal in shortest distance as the success ratio. The results are shown in Table 5b for the best agents throughout the training process.

As in this simple task with relative small state space, DQN gets some performance due to running an average policy across all the the goals, but this is not addressing the task we set out to do. In contrast, GDQN parametrized specifically to include goal information achieves significant better results on both sizes of the environment.

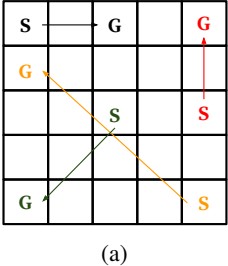

(a)

| Grid Size | DQN | GDQN |
|:---:|:---:|:---:|
| $5 \times 5$ | 0.67 | 0.97 |
| $7 \times 7$ | 0.67 | 0.95 |

(b)

Figure 5: a: Navigation task in gridworld. Each color denotes a different episode, for each episode, a random pair of starting and goal location are generated, the agent needs to reach the goal. b: Results from navigation task.

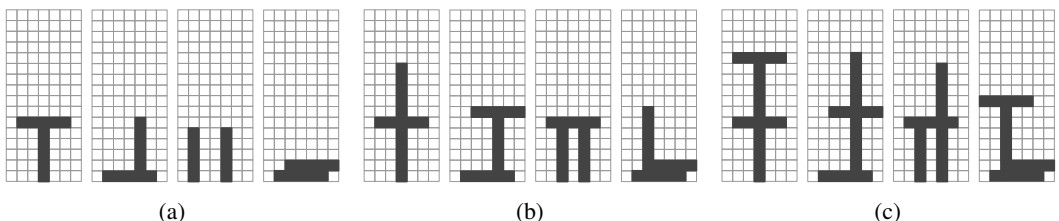

(a)                                    (b)                                    (c)

Figure 6: a: Targets for 2 blocks. b: Targets for 3 blocks. c: Targets for 4 blocks.

## 5.2 TARGET STACKING

We set up 3 groups of target structures consisted of different number of blocks $\{2, 3, 4\}$ in the scene as shown in Figure 6. Within each group of target shapes, a random target (with the accompanied orientation order) is picked at the very beginning for individual episode. Each training epoch consists of 10000 steps and each test epoch with 1000 steps. Similar to the setting in the toy example, all the agents run for 100 epochs and the $\epsilon$ anneals for the first 20 epochs, and the memory buffer size is set as long as the annealing steps at $200K$ steps.

We computed both average overlap ratio (OR) and success rate (SR) for the finished stacking episodes in each test epoch. Here overlap ratio is the same as defined in the reward shaping in Equation 4, but simply measures the end scene over the assigned target scene. This tells the relative completion of the stacked structure in comparison to the assigned target structure, the higher the value is, the better completion it is to the target. At the maximum of 1, it suggests completely reproduction of the target. The success rate counts the ratio how many episodes complete the exact same shape as assigned over the total number of episodes finished in the test epoch. This is the absolute metric counting overall successful stacking. The results are shown in Table 1 for the best agents throughout the training process.

Over all groups on both metrics, we observe GDQN outperforms DQN, showing the importance of integrating goal information. In general, the more blocks in the task, the more difficult it becomes. When there are only small number of blocks (2 blocks and 3 blocks) in the scene, the single policy learned by DQN averages over the few target shapes can still work to some extent. However when introducing more blocks into the scene, it becomes more and more difficult for this averaged model to handle. As we can see from the result, there is already a significant decrease of performance (success rate drops from 0.70 to 0.43) when increasing the blocks number from 2 to 3, whereas GDQN's performance only decreases slightly from 0.72 to 0.67. In 4 blocks scene, the DQN can no longer reproduce any single target (0.0 for success rate, 0.03 for overlap ratio) while GDQN parametrized specifically to include goal information can still do. Though the success rate (absolute completion to the target) for the basic GDQN is relatively low at 0.17 but the average overlap ratio (relative completion to the target) still holds up pretty well at 0.41. Also we see reward shaping can further improves GDQN model, in particular distance transform can boost the performance more than overlap ratio.

| Num. of Blks. | DQN | | GDQN | | GDQN + OR | | GDQN + DT | |
|---|---|---|---|---|---|---|---|---|
| | OR | SR | OR | SR | OR | SR | OR | SR |
| 2 | 0.70 | 0.70 | 0.82 | 0.72 | 0.84 | 0.77 | **0.88** | **0.78** |
| 3 | 0.43 | 0.43 | 0.76 | **0.67** | **0.86** | 0.63 | 0.83 | 0.65 |
| 4 | 0.03 | 0.0 | 0.41 | 0.17 | 0.73 | 0.55 | **0.79** | **0.56** |

Table 1: Results for target stacking. For "GDQN + X", X denotes different ways for reward shaping as described in previous section, OR for overlap ratio, DT for distance transform. For metrics, OR stands for average overlap ratio, SR for average success rate.

## 6  CONCLUSION

We create a synthetic block stacking environment with physics simulation in which the agent can learn block stacking end-to-end through trial and error, bypassing to explicitly model the corresponding physics knowledge. We introduce a target stacking task where the agent stacks blocks to reproduces a tower shown in an image. The task presents a distinct type of challenge requiring the agent to reach a given goal state that may be different for every new trial. Therefore we propose a goal-parametrized GDQN model for stacking to plan with respect to the specific target structure, allowing better generalization across different goals. We validate the model on both a navigation task in a classic grid-world environment with different start and goal positions and the block stacking task itself with different target structures.

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
