# OpenReview forum: "Acquiring Target Stacking Skills by Goal-Parameterized Deep Reinforcement Learning"
_ICLR.cc/2018/Conference — Reject_

### Official Review · AnonReviewer2 · 2017-11-27
**Despite achieving better results than a state of the art method, the presented evaluation is weak and do not convince about the claim (make agents understand physical interaction).**

**Rating:** 5
**Confidence:** 4

**Review:**

The authors propose a model for learning physical interaction skills through trial and error. They use end-to-end deep reinforcement learning - the DQN model - including the task goal as an input in order to to improve generalization over several tasks, and shaping the reward depending on the visual differences between the goal state and the current state. They show that the task performance of their model is better than the DQN on two simulated tasks.
The paper is well-written, clarity is good, it could be slightly improved by updating the title "Toy example with Goal integration" to make it consistent with the naming "navigation task" used elsewhere.

If the proposed model is new given the reviewer's knowledge, the contribution is small. The biggest change compared to the DQN model is the addition of information in the input.
The authors initially claim that "In this paper, [they] study how an artificial agent can autonomously acquire this intuition through interaction with the environment", however the proposed tasks present little to no realistic physical interaction: the navigation task is a toy problem where no physics is simulated. In the stacking task, only part of the simulation actually use the physical simulation result. Given that machine learning methods are in general good at finding optimal policies that exploit simulation limitations, this problem seems a threat to the significance of this work.

The proposed GDQN model shows better performance than the DQN model. However, as the authors do not provide in-depth analysis of what the network learns (e.g. by testing policies in the absence of an explicit goal), it is difficult to judge if the network learnt a meaningful representation of the world's physics. This limitation along with potential other are not discussed in the paper.

Finally, more than a third (10/26) references point to Arxiv papers. Despite Arxiv definitely being an important tool for paper availability, it is not peer-reviewed and there are also work that are non-finished or erroneous. It is thus a necessary condition that all Arxiv references are replaced by the peer-reviewed material when it exist (e.g. Lerer 2016 in ICML or Denil 2016 in ICLR 2017), once again to strengthen the author's claim.

---

### Official Review · AnonReviewer1 · 2017-11-27
**Good performance on an interesting physics task, but the model is not novel**

**Rating:** 4
**Confidence:** 4

**Review:**

Summary: This paper proposes to use deep Q-learning to learn how to reconstruct a given tower of blocks, where DQN is also parameterized by the desired goal state in addition to the current observed state.

Pros:
- Impressive results on a difficult block-stacking task.

Cons:
- The idea of parameterizing an RL algorithm by goals is not particularly novel.

Quality and Clarity:

The paper is extremely well-written, easy to follow, and largely technically correct, though I am somewhat concerned about how the results were obtained as it does not seem like the vanilla DQN agent could do so well, even on the 2-block scenes. Even just including stable scenes, I estimated based on Figure 5 that there must be about 70 different configurations that are stable (and this is likely an underestimate). So, if each of these scenes occurs equally often and the vanilla DQN agent does not receive any information about the target goal and just acts based on an "average" policy, I would expect it to only achieve success about 1/70th of the time. Am I missing something here?

Another thing that was unclear to me is how the rotation of the blocks is chosen: is the agent given the next block with the correct rotation, or can it also choose to rotate the block? In the text it is implied that the only actions are {left, right, down}, which seems to simplify the task immensely. It would be interesting to include results where the agent additionally has to choose from actions of {rotate left by 90 degrees, rotate right by 90 degrees}.

Also: are the scenes used during testing separate from those used during training? If not, it's not obvious that the agent isn't just learning to memorize the solution (which somewhat defeats the idea behind parameterizing the Q-network with new goals every time).

Originality and Significance:

The block-stacking task is very cool and is more complex than many other physics-based RL tasks in the literature, which often involve just stacking square blocks in a single tower. I think it is a useful contribution to introduce this task and the GDQN agent as a baseline. However, the notion of parameterizing the policy by the goal state is not particularly novel. While it is true that many RL papers do train to optimize just a single reward function for a single goal, it is also very straightforward to modify the state space to include a goal and indeed [1-4] are just a few examples of recent papers that have done this. In general, any time there is a procedurally generated environment (e.g. Sokoban, as in [5]) the goal necessarily is included as part of the state space---so the idea of GDQN isn't really that new.

[1] Oh, J., Singh, S., Lee, H., & Kohli, P. (2017). Zero-Shot Task Generalization with Multi-Task Deep Reinforcement Learning. arXiv Preprint arXiv:1706.05064.
[2] Dosovitskiy, A., & Koltun, V. (2017). Learning to act by predicting the future. Proceedings of the 5th International Conference on Learning Representations (ICLR 2017).
[3] Hamrick, J. B., Ballard, A. J., Pascanu, R., Vinyals, O., Heess, N., & Battaglia, P. W. (2017). Metacontrol for adaptive imagination-based optimization. Proceedings of the 5th International Conference on Learning Representations (ICLR 2017).
[4] Pascanu, R., Li, Y., Vinyals, O., Heess, N., Buesing, L., Racanière, S., … Battaglia, P. (2017). Learning model-based planning from scratch. arXiv Preprint arXiv: 1707.06170. Retrieved from https://arxiv.org/abs/1707.06170
[5] Weber, T., Racanière, S., Reichert, D. P., Buesing, L., Guez, A., Rezende, D. J., … Wierstra, D. (2017). Imagination-Augmented Agents for Deep Reinforcement Learning. arXiv Preprint arXiv: 1707.06203. Retrieved from http://arxiv.org/abs/1707.06203

---

### Official Review · AnonReviewer3 · 2017-11-30
**Nice goal augmentation in state representation for DQN with, unfortunately incomplete and quite preliminary**

**Rating:** 5
**Confidence:** 3

**Review:**

The authors use a variant of deep RL to solve a  simplified 2d physical stacking task. To accommodate different goal stacking states the authors extend the state representation of DQN. The input to the network is the current state of the environment as represented by the 2d projection of the objects in the simulated grid world and a representation of the goal state in the same projection space. The reward function in its basic form rewards only the correctly finished model. A number of heuristics are used to augment this reward function so as to provide shaping rewards along the way and speed up learning. The learnt policy is evaluated on the successful assembly of the target stack and on a distance measure between the stack specified as goal and the actual stack.

Currently, I don’t understand from the manuscript, how DQN is actually trained. Are all different tasks used on a single network? If so, is it surprising that the network performs worse than when augmenting the state representation with the goal? Or are separate DQNs trained for multiple tasks?

The definition of value function at the bottom of page 4 uses the definition for continual tasks but in the current setting the tasks are naturally episodic. This should be reflected by the definition.

It would be good if the authors could comment on any classic research in RL augmenting the state representation with the goal state and any recent related developments, e.g. multi-task RL or the likes of Dosovitskiy & Koltun “Learning to act by predicting the future”.

It would be helpful do obtain more information about the navigation task, especially a plot of sorts would be helpful. Currently, it is particularly difficult to judge exactly what the authors did.

How physically “rich” is this environment compared to some of the cited work, e.g. Yildirim et al. or Battaglia et al:?

Overall it feels as if this is an interesting project but that it is not yet ready for publication.

---

### Decision · Program_Chairs · 2018-01-29
**ICLR 2018 Conference Acceptance Decision**

**Decision:**

Reject

**Comment:**

The authors present a toy stacking task where the goal is to stack blocks to match a given configuration, and a method that is a slightly modified DQN algorithm where the target configuration is observed by the network as well as the current state. There are a few problems with this paper. First, the method lacks novelty - it is very similar to DQN. Second, the claims of learning physical intuitions is not borne out by the method or experimental results. Third, the tasks are very simple and there is no held-out test set of target configurations.